# Taloe—Sedimentation in an Intermittent Lake (Russian Federation, Republic of Khakassia)

**Maria O. Khrushcheva [1,\*], Ekaterina M. Dutova [1,2], Platon A. Tishin [1], Alexander L. Arkhipov [1], Alexei N. Nikitenkov [2] and Alexei I. Chernyshov [1]**

[1] Department of Geology and Geography, Tomsk State University, Prospekt Lenina 36, 634050 Tomsk, Russia; tishin_pa@mail.ru (P.A.T.); alexlarhipov@gmail.com (A.L.A.); aich@ggf.tsu.ru (A.I.C.)

[2] Department of Natural Resources, Tomsk Polytechnic University, Prospekt Lenina 30, 634050 Tomsk, Russia; dutova@tpu.ru (E.M.D.); corestone@mail.ru (A.N.N.)

\* Correspondence: masha2904@mail.ru

**Abstract:** This paper examines the mineral and geochemical features of lake sediments and waters in intermittent Lake Taloe, located in a semiarid climate. Minerals that belong to groups of oxides, sulfides, aluminosilicates, carbonates, sulfates, and halides are identified through the use of precision methods. The resulting mineral species are divided by genetic features into two associations: terrigenous and hydrogenic. The terrigenous association includes water-insoluble minerals, while the hydrogenic association combines typical hydrogenic minerals. The regularities of the accumulation and distribution of minerals along the lake laterally and to a depth of up to one meter are also examined. The order of deposition of hydrogenous association minerals from sulfate-chloride lake waters was established. The obtained results are confirmed and supplemented by physicochemical calculations, which show the equilibrium of lake waters with hydroxides, oxides, aluminosilicates, carbonates, and sulfates. It has been established that the formation of minerals mainly occurs through evaporative concentration in conjunction with bedrock weathering.

**Keywords:** X-ray diffractometry; salt lakes; intermittent lakes; mineral formation; evaporative concentration; South Minusinsk Basin

## 1. Introduction

Many studies have been devoted to the problems of studying salt lakes [1–9]. The formation of various lakes, factors of formation of evaporite sediments, and salt deposits are often discussed [3,10–13]. There are works devoted to the mineralogy and geochemistry of salts in salt lakes, the results of which are used to assess the evaporation rate and study the evolution of brines and paleoclimate features [14–19]. The most relevant publications are devoted to the research of sediment formation processes in lake systems as well as works reflecting the latest methods and approaches in the study of such processes [20,21]. Lake systems associated with orogenic depressions of the steppe, desert, and semi-desert landscapes are often included in these works. Various studies are dedicated to the different types of the world's salt lakes: soda, sulfate, chloride types. There are lakes in Mongolia, China, Iran, Turkey, India, etc. among the examples [7,8,14,15,17]. There are specific means of component concentration, sources, and geochemical characteristics for each type. In Russia, a striking example of such territories is the South Minusinsk Basin (Republic of Khakassia), this steppe region contains a large concentration of mineralized intermittent lakes [10,22–26].

In this paper, the mineralogical and geochemical features of the waters and lake sediments in Taloe, an intermittent lake located in the steppe zone with a predominantly semiarid climate, are examined. Taloe lake waters are of sulfate–chloride type, for which high salinity and slightly alkaline pH are typical.

The primary objective of this work is to study the nature of mineral species formation in sulfate-chloride intermittent lakes in a semiarid climate and to calculate the physico-chemical equilibria of the water-mineral system to confirm the experimental results.

## 2. Characteristics of the Area and Object of Research

The object of the study is lake Taloe, an intermittent body of water; it is the most striking example of such lakes. Moreover, in the first half of the 20th century, the largest salt deposit in the area existed within its basin [27]. Geographically, the Lake Taloe watershed belongs to the Uibat steppe, located in the north-eastern part of the South Minusinsk Basin in the Republic of Khakassia (Figure 1). Within the immediate vicinity are the cities of Chernogorsk and Abakan.

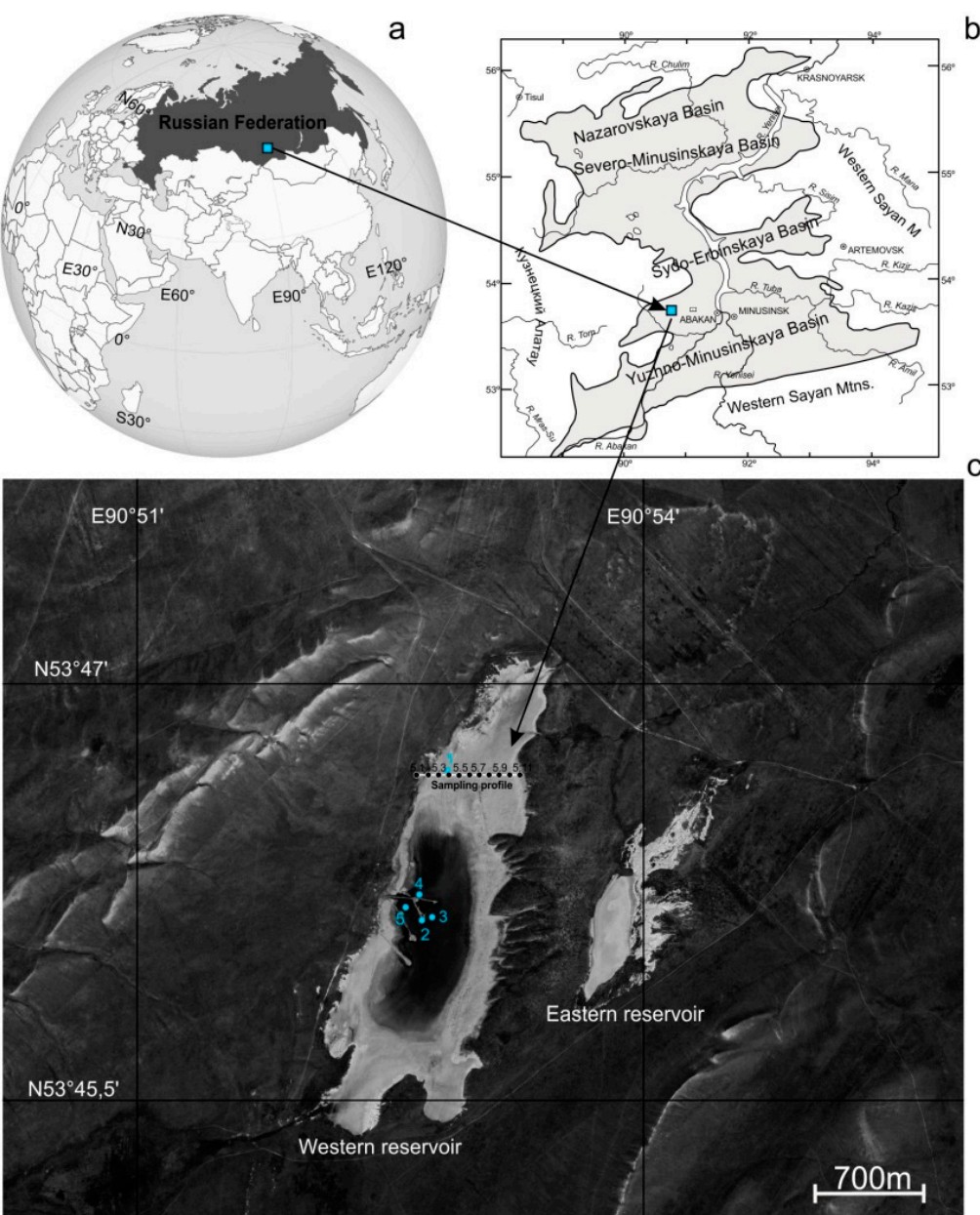

**Figure 1.** Quick map of the location of Lake Taloe with a map of the actual material (**c**). The figure shows the geographic location of Lake Taloe (**a**,**b**). Note: points 5.1–5.11 (black) are lake sediments sampled from a profile 50 m apart. Points 1–5 (blue)—water samples.

The choice for the object of study was determined by its territorial location at a high latitude (53°–54° N), characterized by contrasting changes in solar activity and seasonal temperature fluctuations. The average annual air temperature varies from 1.8 °C to minus 1.0 °C. The coldest temperature on record (minus 43.8 °C) was recorded in December 1966, and the hottest on record (36.0 °C) in July 1970. The climate of this area is semiarid; it is characterized by arid, hot summers and quite harsh winters. A significant feature of the area is its active aeolian processes (wind speed up to 20–30 m/s), contributing to the transfer of detrital material into the lake basin.

The northwestern part of the South Minusinsk basin is located within the steppe zone; it is a forestless plain with cuesta-like hills and ridges from 400 to 520 m high (rarely higher). From the central part of the study area, in the north-east and south-west directions, the topography is calmer with relative elevations of 430–470 m, rarely more. The relative elevation of the terrain varies within 130 m. The terrain exposure is medium; the thickness of loose deposits varies from 0.5 to 2.5 m, reaching in the lower parts of the terrain 5–10 m. Unconsolidated sediments of northwest part of Minusinsk basin formed in Eopleistocene and early Neo-Pleistocene.

Lake Taloe is a lake-chemogenic sub-plain filling a depression within the denudation plain located between the remnants of the cuesta hillocky area. Underlying rocks are carbonate-terrigenous sedimentary rocks of Late Devonian Tubinskaya suite. Morphologically, Lake Taloe consists of two intermittent lakes (playa) with dimensions of minerals-12001453300 m × 960 m and 1610 m × 385 m, elongated in the meridional direction, which are filled with water during periods of intense precipitation (snowmelt, precipitation). Conventionally, these two lakes are designated Eastern and Western (Figure 1).

The primary feed in the lakes is due to underground water. Lake sediments of the studied lakes are similar in structure to hydromorphic salt marshes. At a depth of 0–10 cm, the horizon is composed of brown sediments that differ slightly from the underlying layer. The sediments from the immediate surface are covered by a salt crust with a thickness of about 2–3 cm, represented by a loose mass of soil particles and salt crystals.

In the range 10–50 cm, there is a humus horizon, of uniform composition, with glaucous, gleic and bluish-green interlayers, which remain until the end of the section. At a depth of 50–90 cm, the mineral composition is similar to the interval of 10–50 cm, there are intensively gleic (blue-gray) areas. At a depth of 90–100 cm, there is a typical salt marsh gleic bedrock with the characteristic smell of hydrogen sulfide (Figure 2).

Taloe lake sediments themselves both in shore and central parts formed in Holocene. According to [28], the average sedimentation rate at the Minusinsk basin lakes is about 2 mm per year. The sediments were dated by Rogozin D. Yu. [28] based on [137]Cs, [210]Pb and [14]C isotopes study [28]. Considering that sampling was done at ≤1 m deep, the sediments analyzed within the present study formed in the last 500 years.

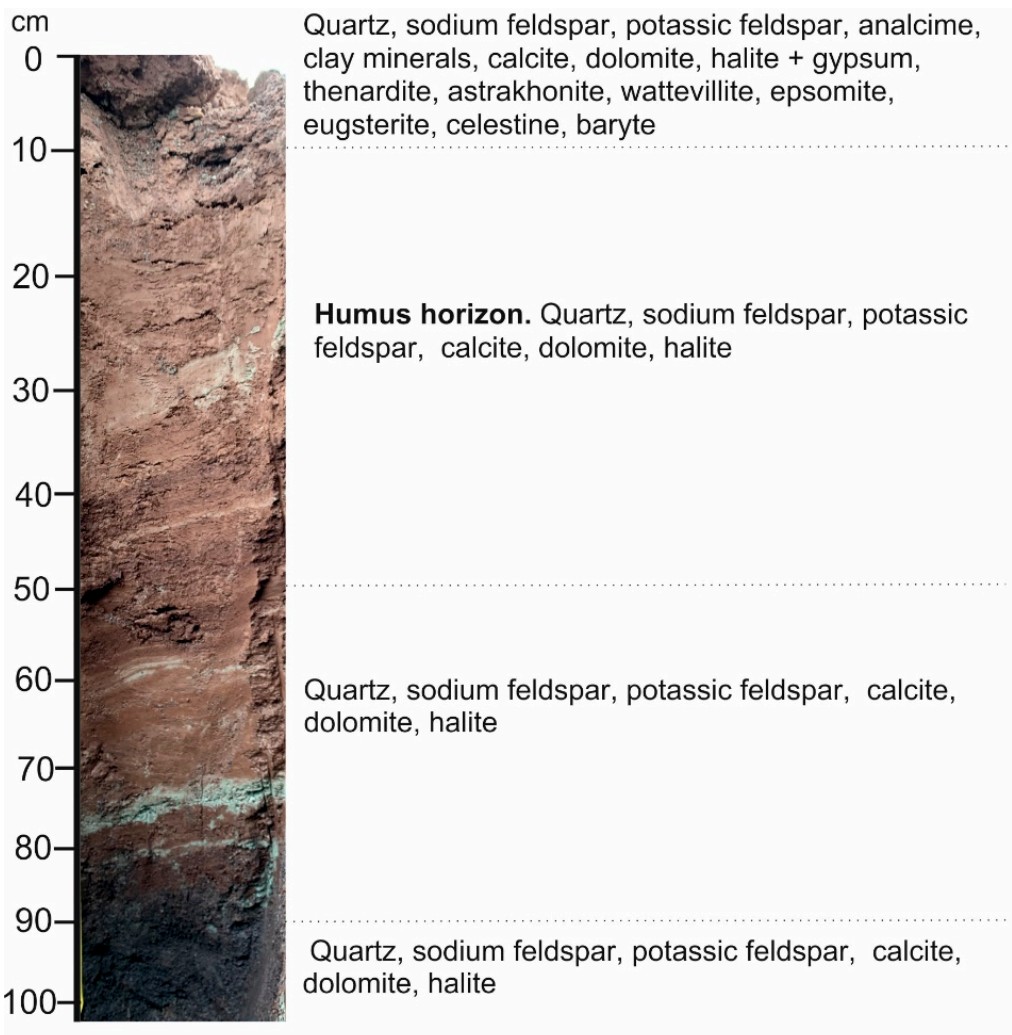

**Figure 2.** Distribution of minerals in the sediments of Lake Taloe in the section.

## 3. Materials and Methods

### 3.1. Sampling

The material of solid lake sediments for research was selected during the dry periods of 2018–2019. Sampling was carried out using a drill with a diameter of 12 cm along the profile in a west–east direction. Sampling points were located 50 m from each other, and in each of the points, 10 samples were taken from the surface to a depth of 1 m (every 10 cm). A total of 110 samples were selected. The position of the section under study is shown in Figure 2.

Lake water samples and samples of pore waters of lake sediments of the lake were taken directly from the lake surface (samples 3, 4) to study the geochemical features and mineral-forming ability of water media. The latter was taken from a technical well with a depth of 8 m (sample 2), located directly in the area of the lake under study, and from a pit with a depth of 1 m (sample 1). To characterize the underground waters of the Upper Devonian aquifer complex, the materials of previously conducted studies were used [22].

### 3.2. Determination of the Chemical Composition of Solid Sediments and Water

Determination of the content of micro-components in water and lake sediments was performed by inductively coupled plasma mass spectrometry on the Agilent 7500cx, Agilent Technologies device at the accredited laboratory for the Central Research Center of "Analytical Center for Geochemistry of Natural Systems" at Tomsk State University, according to the method described in [29]. The analytical work quality was controlled

based on the HighPurity CRM-TMDW Standard for Drinking Water (water samples) and GSO 3483-86 SGHM-1 (lake sediment) standards.

The determination of water components (anions and cations) was carried out by complex titrimetry ($HCO_3$, $CO_2$, $CO_3$, $Cl^-$, $Ca^{2+}$, $Mg^{2+}$), flame photometry (Na, K), potentiometry (pH), and turbidimetry ($SO_4^{2-}$) at the accredited Problem Research Laboratory of Hydrogeochemistry at Tomsk Polytechnic University.

### 3.3. Total Mineral Composition

The study of the total mineral composition of lake sediments was carried out by X-ray diffractometry. The analysis was carried out at the collective use center "Analytical Center of Geochemistry of Natural Systems" of Tomsk State University on the X'pertPro X-ray Diffractometer from the PANalytical company (the Netherlands). Case: 40 kV, 30 mA, Cu radiation, Ni filter, measuring range 3–60° 2θ. The diffractograms were decoded using the HighScore software and the PDF-4 Minerals 2021 database (ICDD company).

Minerals occurring in negligible concentrations were analyzed based on an analytical complex of a scanning electron microscope MIRA 3 LMU combined with an energy-dispersive X-ray spectrometer of the UltimMax 40 model.

**Clay Minerals Analysis**

The clay minerals analysis was carried out by X-ray diffractometry in a fraction smaller than 10 μm. This fraction was obtained by the precipitation of the starting material in distilled water. The precipitation time was calculated using Stokes' formula [30]. The resulting fraction was divided into three additional samples: one remained untreated, the second sample was saturated with ethylene glycol vapor, and the third sample was calcined in a muffle furnace at 550 °C for 2 h. Oriented preparations were prepared from the obtained samples.

The identification of clay minerals was based on the position of the *001* series of basal reflections and the comparison of the three states of the samples mentioned above.

The presence of easily soluble salts (mainly halite) in the samples was complicated in the clay minerals identification, which was reflected in the diffractograms in the form of intense reflections, against which the reflections of clay minerals looked faint.

### 3.4. Physical and Chemical Calculations

The authors carried out physicochemical calculations for the equilibrium of water with major minerals, including hydrogeochemical testing of the probability of one or another association of hydrogenous mineral formation to clarify the genesis of the formed sediments.

As a tool for studying the mineral-forming ability of water media, the software package HydroGeo (HydroGeo version 04022021, Michael Bukaty, Aleksei Nikitenkov at Tomsk Polytechnic University, Tomsk, Russia) developed by Bukaty was selected [31–33]. The HydroGeo PC, similar to many other similar models developed in Western Europe or the United States (for example, MINTEQA (version 2, Jon Petter Gustafsson at KTH, Stockholm, Sweden) or PHREEQC (version 3, Richard Webb at USGS Water Resources Mission Area, Reston, Virginia, USA)) [34,35], simulates thermodynamic processes of dissolution and precipitation, solving the problems of minimizing free energy. HydroGeo relies on an extensive database of thermodynamic data, which contains characteristics of minerals, water ions, and complex compounds, taken from various sources, including cprons92, SUPCRT (CPRONS92, SUPCRT92 thermodynamic databases contain parameters for the calculation of standard thermodynamic properties of aqueous species, gases, and minerals for temperatures 0–1000 °C and pressures 1–5000 bar), and NAGRANTB (thermodynamic data from reports) report series [36–38].

The HydroGeo model has been extensively tested on comparable international models, including the HMV model [33,39]. The use of the HydroGeo software for solving a wide range of different tasks has been previously demonstrated [40–46]. The mineral-forming

capacity of fresh underground waters and their corresponding new mineral formations were studied with the utmost care and attention to detail [47–49].

Physical and chemical calculations for the study of the processes of Lake Taloe were carried out in relation to the average landscape and climatic conditions of the area (T = 10 °C, Pgen = 1 atm). The system was characterized by a matrix of 20 base ions, 102 components of an aqueous solution, and 54 minerals.

The equilibrium of water with minerals was assessed by calculating the water-rock system indices (disequilibrium indices). The positive values of the disequilibrium indices obtained by the calculation indicate the ability of the system to form certain minerals, and the negative values indicate the ability of the system to dissolve them.

## 4. Results and Discussion

### 4.1. Mineralogy of Lake Sediments

As a result of X-ray tests conducted on the sediments of Lake Taloe, the following mineral species were identified: the main sedimentary minerals are quartz (up to 37%), clay minerals are represented by minerals of the chlorite group, illite, and mixed-layer formations (MLF) of the illite-montmorillonite series (up to 46%) and calcite (up to 29%), less common are feldspars (albite (up to 15%) and potassic feldspar (up to 7%)), gypsum (up to 13%) and halite (up to 14%), in low concentrations, there is dolomite (up to 5%), analcime (up to 4%), aragonite, pyrite, hematite, thénardite, blödite, wattevillite, epsomite, eugsterite, celestine, and baryte (less than 1%).

The results of the mineral composition, according to X-ray diffractometry, are shown in Table 1.

**Table 1.** Generalized data on the mineral composition of Lake Taloe sediments (according to X-ray diffractometry), %.

| Group of Minerals | Mineral | Chemical Formula of the Mineral | Coastal Zone | | | Central Zone | | |
|---|---|---|---|---|---|---|---|---|
| | | | Avg | Min | Max | Avg | Min | Max |
| Oxides | Quartz | $SiO_2$ | 30 | 25 | 35 | 28 | 22 | 37 |
| | Hematite | $Fe_2O_3$ | <1 | <1 | <1 | <1 | <1 | <1 |
| Sulfides | Pyrite | $FeS_2$ | <1 | <1 | <1 | <1 | <1 | <1 |
| Aluminosilicates | Chlorite group | $(Mg, Fe, Al, Cr, Ni, Mn)_3(Si, Al)_4$ $O_{10}(OH)_2 \times (Mg, Fe, Mn)_3(OH)_6$ | 17 | 12 | 22 | 15 | 6 | |
| | Illite | $(K_{0.75}(H_3O)_{0.25})Al_2(Si_3Al)$ $O_{10}((H_2O)_{0.75}(OH)_{0.25})_2$ | 14 | 10 | 19 | 16 | 9 | 32 |
| | MLF [1] | - | 5 | 2 | 9 | 4 | 2 | 7 |
| | Albite | $Na[AlSi_3O_8]$ | 10 | 6 | 15 | 10 | 5 | 15 |
| | Potassic feldspar | $K[AlSi_3O_8]$ | 2 | 0 | 3 | 2 | 0 | 7 |
| | analcime | $Na[AlSi_2O_6] \times H_2O$ | 1 | 0 | 2 | 2 | 0 | 4 |
| Carbonates | Calcite | $CaCO_3$ | 17 | 10 | 25 | 19 | 10 | 29 |
| | Dolomite | $CaMg(CO_3)_2$ | 2 | 0 | 4 | 2 | 0 | 5 |
| | Aragonite | $CaCO_3$ | <1 | <1 | <1 | <1 | <1 | <1 |
| Sulfates | Gypsum | $CaSO_4$ | n.f. | n.f. | n.f. | 6 | 0 | 13 |
| | Thénardite | $Na_2SO_4$ | n.f. | n.f. | n.f. | <1 | 0 | <1 |
| | Blödite | $Na_2Mg(SO_4)_2 \times 4H_2O$ | n.f. | n.f. | n.f. | <1 | 0 | <1 |
| | Wattevillite | $Na_2Ca(SO_4)_2 \times 4H_2O$ | n.f. | n.f. | n.f. | <1 | 0 | <1 |
| | Epsomite | $MgSO_4 \times 7H_2O$ | n.f. | n.f. | n.f. | <1 | 0 | <1 |
| | Eugsterite | $Na_4Ca(SO_4)_3 \times 2H_2O$ | n.f. | n.f. | n.f. | <1 | 0 | <1 |
| | Celestine | $SrSO_4$ | n.f. | n.f. | n.f.. | <1 | 0 | <1 |
| | Baryte | $BaSO_4$ | n.f.. | n.f. | n.f. | <1 | 0 | <1 |
| Haloids | Halite | $NaCl$ | 5 | 0 | 11 | 5 | 0 | 14 |

[1] MLF—mixed-layer formations of the illite-montmorillonite series. n.f.—not found. – indicate the formula is missing.

### 4.1.1. Oxides, Sulfides, and Aluminosilicates

Quartz is identified by a series of basic basal reflections *hkl* ($d_{intensity}$): *011* ($3.34_{10}$), *100* ($4.25_3$), *112* ($1.81_2$) Å. Quantitatively, its content varies from 25 to 35%, with an average concentration of 30%. In the direction from the shoreline to the central part of the lake and as it deepens, its amount varies slightly, and in general, one can indicate its relatively uniform distribution in the territory of the studied lake. Quartz is considered as one of the main terrigenous minerals, while the possibility of its partially hydrogenous origin is not fully excluded [50].

Hematite was identified by characteristic reflections *hkl* (d, intensity): *104* ($2.69_{10}$), *110* ($2.51_8$), *024* ($1.84_6$) Å. The hematite concentrations in sediments are negligible and are quantitatively expressed in a content of less than 1%.

Pyrite was identified by a series of interplanar distances *hkl* ($d_{intensity}$): *200* ($2.70_{10}$), *111* ($3.12_4$), *210* ($2.42_6$) Å. Its concentrations are noted at the level of trace amounts (less than 1%). Pyrite is distributed throughout the entire area of the lake. The presence of hydrogen sulfide in the lake creates a favorable environment for the formation of iron sulfide. Such processes with sulfur minerals study are described in detail for Transbaikalian salt lakes by Borzenko S. V. [51].

The mineral of the zeolitic group is represented by analcime. It is determined by the characteristic set of reflections *hkl* ($d_{intensity}$): *404* ($3.42_{10}$), *122* ($5.61_8$), *054* ($2.92_7$) Å. Analcime was identified in single samples; its maximum content (4%) was recorded at the sampling point located 350 m from the lake shoreline. Presumably, analcime can be formed due to the alteration of sodium plagioclase in the environment of underground water migration.

Feldspars are represented by albite; to a lesser extent by potassic feldspar. Albite was identified by diffraction lines *hkl* ($d_{intensity}$): *002* ($3.18_{10}$), *$\overline{2}01$* ($4.03_8$), *$\overline{1}30$* ($3.69_4$), *$\overline{2}02$* ($3.21_7$) Å, potassic feldspar by reflections *002* ($3.23_{10}$), *220* ($3.33_6$), *$\overline{11}2$* ($3.46_5$), *$\overline{1}30$* ($3.75_5$) Å. Sodium feldspar is characterized by a relatively uniform distribution in sediments and is distributed over the entire area of the lake and over the entire studied depth from 5 to 15%. Potassic feldspar is less common and is found in isolated samples. Its concentrations vary from 1 to 7%. The accumulation of potassic feldspar is characteristic mainly for the lower parts of the studied sediments (50–100 cm from the surface). Feldspars are terrigenous minerals, the accumulation of which occurs from the nearest catchment area; however, the literature notes the possibility of the formation of hydrogenated albite in salt lakes [51]. According to physicochemical calculations, formation of hydrogenous albite at Taloe lake is possible at relatively high pH (>7.5) and mineralization (>85 g/L).

Clay minerals are represented by three varieties: minerals of the chlorite group (iron-magnesian chlorite), illite, and mixed-layer formations of the illite-montmorillonite series (MLF). X-ray diffraction of iron-magnesia chlorite was determined by a series of reflections *hkl* ($d_{intensivity}$): *001* ($14.10_3$), *002* ($7.05_{10}$), *003* ($4.70_2$), *004* ($3.53_7$) Å. Iron-magnesia chlorites are characterized by an increased intensity of *003*, close to the reflections *002* and *004*, and after sample calcination, the chlorite reflections are preserved. Minerals of the chlorite group differs from kaolinite by the position of the *004* reflex. In this paper, the term "illite" was used according to the Nomenclature Committee's recommendations [52], which implies the group name of all micaceous minerals in which the number of swollen smectite layers does not exceed 15%. X-ray illite identification was performed using a characteristic set of reflections *hkl* ($d_{intensivity}$): *002* ($9.94_{10}$), *004* ($4.97_4$), *006* ($3.35_{10}$), *00 10* ($1.99_4$) Å. When calcining and saturating the sample with ethylene glycol, the interplane distances do not change due to the robust and stable illite trellis. Mixed-layer formations of the illite-montmorillonite series were identified by a characteristic broad diffuse reflex in the region of 11.00–12.10 Å, which, after saturation with ethylene glycol vapor, shifts to a small-angle region up to 16.30 Å. The shift is caused by an increase in the c parameter of the crystal structure of montmorillonite due to the introduction of organic molecules of ethylene glycol–diatomic alcohol $C_2H_4(OH)_2$ into its interlayer gap (Figure 3).

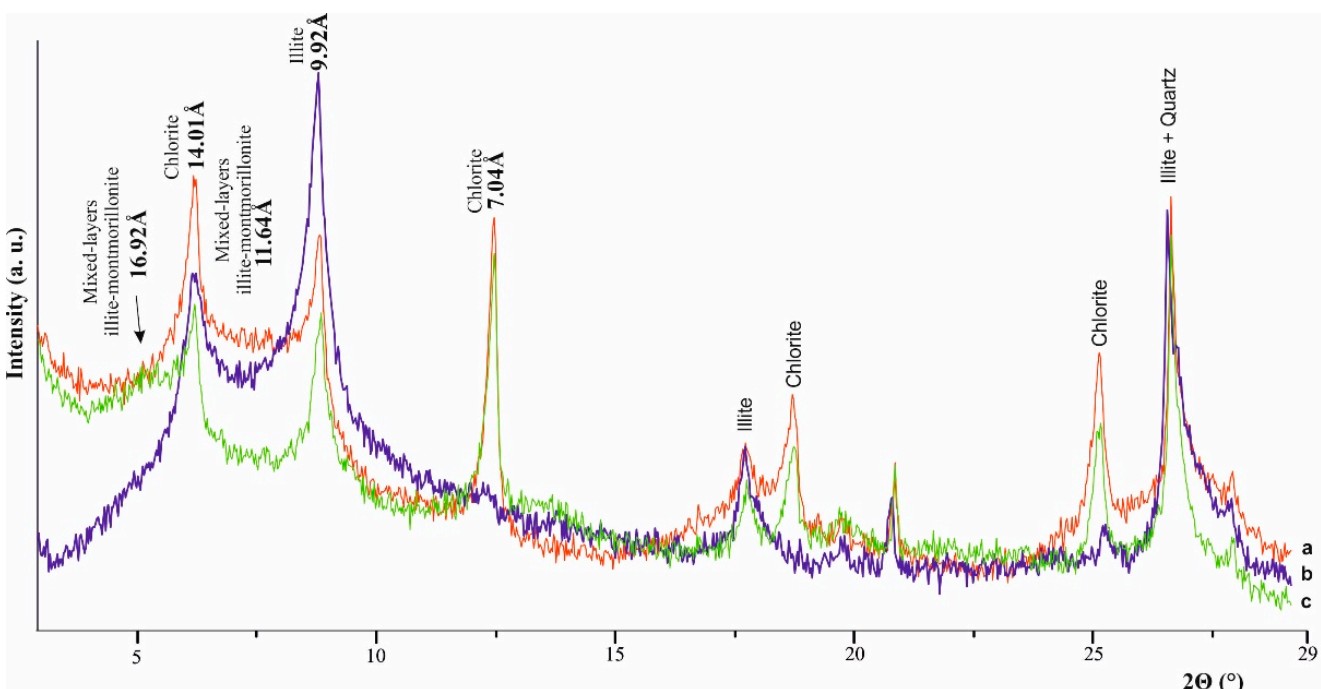

**Figure 3.** Diffractogram of the sample 5.1.20–30 clay component. Red indicates the initial diffractogram (**a**), blue—after calcination at 550 °C (**b**), green—after saturation of the sample with ethylene glycol (**c**).

The share of clay minerals in the total composition of sediments is quite large, from 26 to 46% of the total content of mineral species. Quantitatively, the content of the minerals of the chlorite group in the coastal sediments varies from 12 to 22%, illite from 10 to 19%, and mixed-layer formations from 2 to 9%. In the central part, a slight increase in clay minerals in sediments is observed in quantitative terms: the content of the minerals of the chlorite group from 6 to 28%, illite from 9 to 32%, mixed-layer formations from 2 to 7%. Clay minerals are secondary aluminosilicates that are formed in the zone of groundwater migration when interacting with feldspars [53].

### 4.1.2. Carbonates

The carbonate minerals are represented by calcite, dolomite, and aragonite. Calcite was identified by the main reflections *hkl* ($d_{intensity}$) *104* ($3.03_{10}$), *110* ($2.49_2$), *113* ($2.28_2$), *202* ($2.09_2$) Å, dolomite by peaks *104* ($2.88_{10}$), *012* ($3.69_3$), *006* ($2.66_3$) Å, aragonite by reflections *111* ($3.39_{10}$), *021* ($3.28_5$), *012* ($2.70_5$) Å. The predominant mineral in the carbonates group is calcite; its concentration is from 10 to 29%. The distribution of calcite is fairly uniform, both with depth and laterally across the entire lake. In a subordinate amount, dolomite is found; it is noted in an amount of up to 5%, and the aragonite content is not more than 1%. In the sediments of Lake Taloe, calcite has both hydrogenic and terrigenous origin. This is confirmed by the chemical characteristics of lake waters and the carbonate-terrigenous composition of the underlying rocks. Dolomite and aragonite are typical hydrogenic minerals, which can be formed both due to groundwater interaction with sediments and due to the processes of evaporative concentration of lake water.

### 4.1.3. Sulfates

In Lake Taloe sediments, sulfate minerals were identified: gypsum, thénardite, blödite, wattevillite, epsomite, eugsterite, celestine, and baryte. In the group of sulfates, gypsum is the predominant mineral; it is identified by a series of reflections *hkl* ($d_{intensity}$) $\overline{1}21$ ($4.27_{10}$), *020* ($7.59_8$), $\overline{1}41$ ($3.06_6$), *121* ($2.87_4$) Å. Its content varies from 1 to 5%. The diffractograms also show reflections of epsomite *hkl* ($d_{intensity}$) *211* ($4.21_{10}$), ($4.20_8$), *411*($2.66_3$), and blödite

*210* (4.50$_{10}$), *22$\bar{1}$*(2.93$_4$), *11$\bar{1}$*(2.59$_2$) Å. Other sulfate minerals occur as single grains and are identified by scanning electron microscopy after the concentration of soluble minerals (Figure 4).

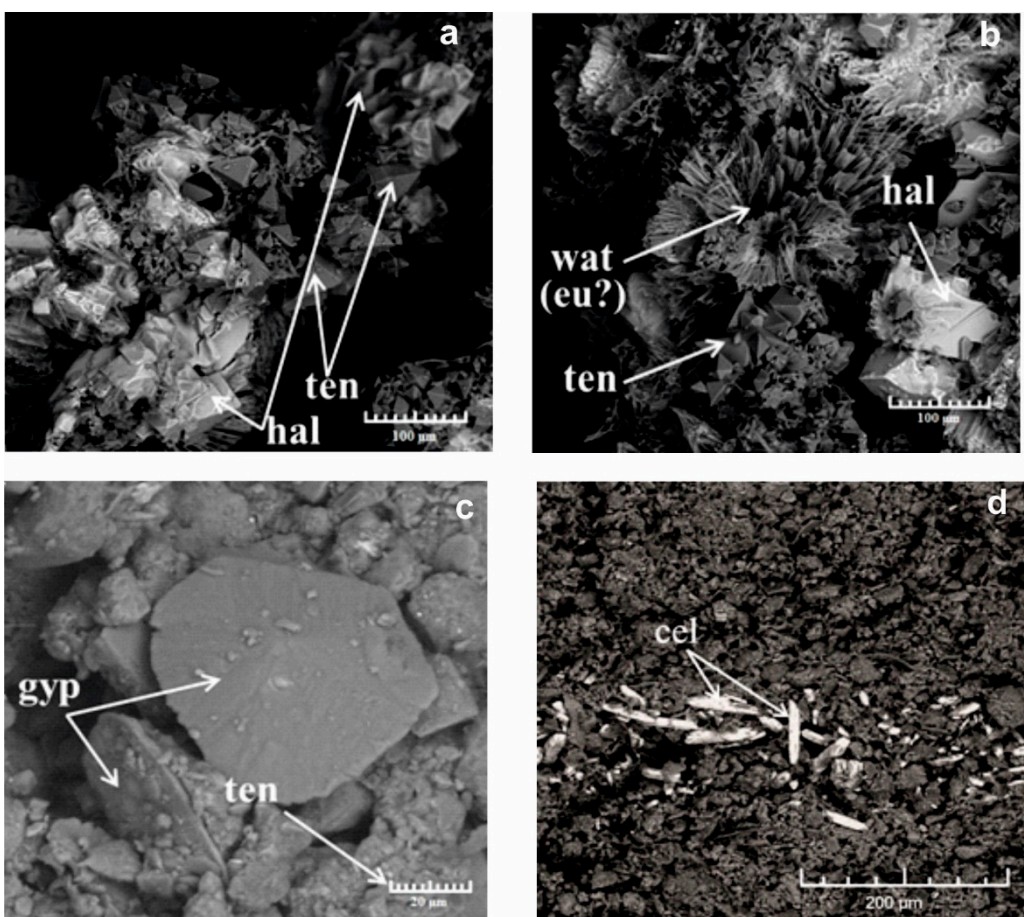

**Figure 4.** Images of the salt crust of lake sediments. (**a–d**): hal—halite, ten—thénardite, wat—wattevillite, eu—eugsterite, gyp—gypsum, cel—celestine.

It is worth noting that sulfate minerals are found exclusively in the upper part of the studied sediments (from the surface to a depth of 20 cm) as part of the salt crust.

### 4.1.4. Haloids

Halite is identified by the characteristic lines *hkl 200* (2.81$_{10}$), *220* (1.99$_6$), *222* (1.62$_2$) Å. It is distributed over the entire area of the lake, but its increased content is recorded at the level of 0–10 cm from the surface of precipitation in the salt crust composition. Its concentrations vary from trace amounts up to 11% in the coastal part and up to 14% in the central part of the lake.

### 4.2. *Geochemical Features and Mineral-Forming Capacity of Underground and Lake Waters*

The chemical compositions of the studied underground, surface, and pore waters of lake sediments are presented in Table 2.

**Table 2.** Chemical composition of Lake Taloe Basin waters and underground waters of the area, mg/L.

| Chemical Composition/Sample | Underground Waters | Water Type | | | | | |
|---|---|---|---|---|---|---|---|
| | | Surface Waters | | | Pore Waters of Lake Sediments | | |
| | Xa-60 | 1 | 2 | 3 | 4 | 5 | 6 |
| pH | 7.27 | 7.58 | 7.37 | 7.27 | 7.60 | 7.47 | 7.41 |
| $HCO_3^-$ | 339 | 244 | 153 | 427 | 183 | 183 | 360 |
| $SO_4^{2-}$ | 456 | 16,230 | 7882 | 10,502 | 19,060 | 20,784 | 8372 |
| $Cl^-$ | 124 | 31,510 | 15,227 | 16,150 | 35,460 | 46,735 | 56,325 |
| Mineralization | 1394 | 75,216 | 36,661 | 42,309 | 86,557 | 10,6916 | 100,293 |
| $Ca^{2+}$ | 120 | 900 | 300 | 460 | 810 | 800 | 963 |
| $Mg^{2+}$ | 92 | 1190 | 378 | 482 | 1202 | 1306 | 1990 |
| $Na^+$ | 143 | 25,009 | 12,650 | 14,214 | 29,204 | 36,937 | 31,700 |
| $K^+$ | 2.3 | 31.4 | 31.2 | 33.2 | 18.0 | 53.0 | 83.2 |
| Br | 0.98 | 92.25 | 38.73 | 40.96 | 97.98 | 118.00 | 200.00 |
| Li | 0.03 | 1.72 | 0.33 | 0.78 | 1.84 | 2.28 | 5.38 |
| Sr | 2.88 | 33.53 | 9.18 | 7.97 | 31.26 | 34.22 | 21.4 |
| B | 0.17 | 8.34 | 13.96 | 3.56 | 8.99 | 11.18 | 1.58 |
| | $\frac{SO_4\ 48\ HCO_3\ 28\ Cl\ 17}{Mg\ 38\ Na\ 31\ Ca\ 30}$ | $\frac{Cl\ 72\ SO_4\ 27}{Na\ 88}$ | $\frac{Cl\ 72\ SO_4\ 28}{Na\ 92}$ | $\frac{Cl\ 67\ SO_4\ 32}{Na\ 91}$ | $\frac{Cl\ 71\ SO_4\ 28}{Na\ 90}$ | $\frac{Cl\ 75}{Na\ 91}$ | $\frac{Cl\ 90}{Na\ 86}$ |

It can be seen from the data that the underground waters of the studied area are of the chloride-bicarbonate-sulfate type of variable cationic composition, with a mineralization of 1.36 g/L and slightly alkaline pH (Figure 5).

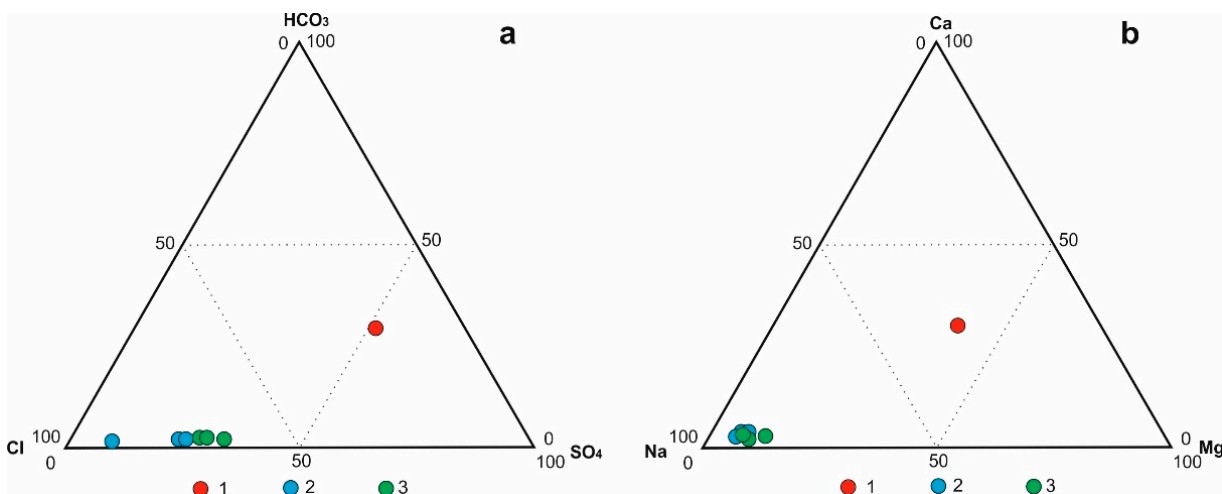

**Figure 5.** Ferre triangles with data on the chemical composition of different types of Lake Taloe waters: groundwater (1), surface water (2), pore water of lake sediments (3). (**a**): anions; (**b**): cations.

The salinity of Lake Taloe water is much higher than that of the underground water. The surface waters of Lake Taloe are of the transitional sulfate–chloride type (Figure 5) with an increased salinity (36.63–75.2 g/L) and a pH value from 7.27 to 7.58. The share of $Cl^-$ accounts for 66–72 g-equ%; in absolute values, its content varies from 15.2 to 31.5 g/L. The amount of $SO_4^{2-}$ varies from 27 to 32 g-equ%. $HCO_3^-$ is present in lower concentrations of 0.3–1 g-equ%. The proportion of $Na^+$ relative to other cations prevails and is 88–92 g-equ%, with absolute contents of 12.6–25 g/L. The proportion of $Mg^{2+}$ varies from 5 to 8 g-equ%. The concentration of $Ca^{2+}$ is 2–3 g-equ%. The $K^+$ content is noted in insignificant amounts and is 0.03 g/L maximum. Pore waters of lake sediments are characterized by the highest salinity and belong to the sulfate–chloride and sulfate types (Figure 5) with mineralization of 86–106 g/L, and the content of $Cl^-$ increases with increasing salinity. $Cl^-$ accounts for 70–75 g-equ% and, in absolute values, 35.4–47.7 g/L. The amount of sulfate varies from 19.0 to 20.7 g/L (24–28 g-equ%). The $HCO^{3-}$ content is at the level of

0.2 g-equ%. The quantitative distribution of cations in the pore waters of lake sediments is similar to their distribution in surface waters. The maximum concentrations are observed for $Na^+$; its content in absolute units is 29.2–36.9 g/L, which corresponds to 90–91 g-equ%. The proportion of $Mg^{2+}$ is 6–7 g-equ%, and $Ca^{2+}$ is about 2 g-equ%. The $K^+$ content in this type of water is very insignificant, up to a maximum of 0.05 g/L.

The analysis of the obtained data allowed for establishing the dependence of the increase in the contents of $Cl^-$, $SO_4^{2-}$ and, on the contrary, the decrease in the concentration of carbonates with increasing mineralization (Figure 6a). The main cations are characterized by an increase in their amount with an increase in the water salinity (Figure 6b).

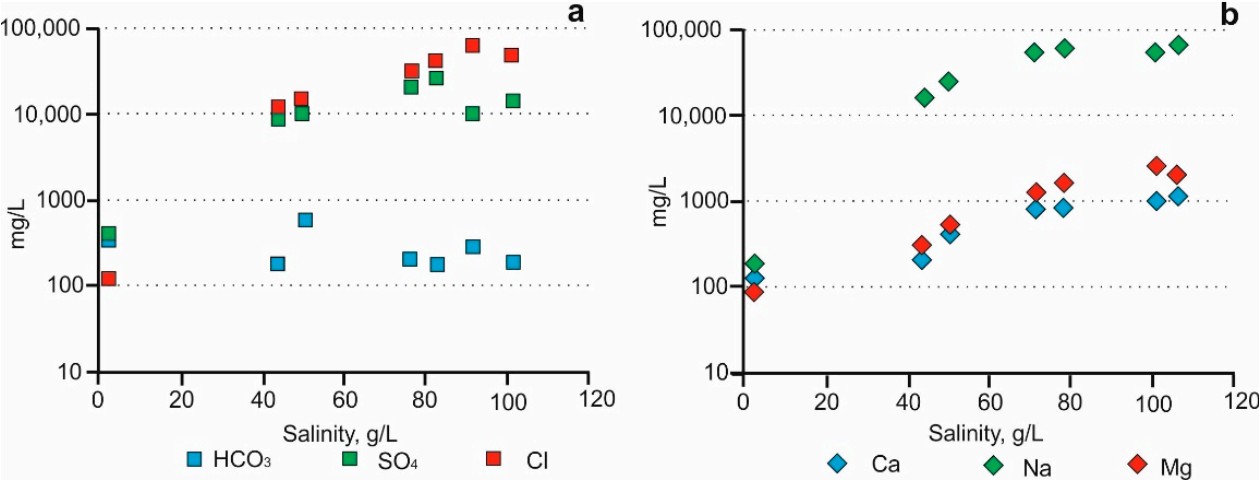

**Figure 6.** Dependence of basic anions (**a**) and cations (**b**) on water mineralization.

Micro-component analysis showed significant concentrations of Br (38.73–118.00 mg/L, with an average value of 77.58 mg/L), Li (0.33–2.28 mg/L, average 1.39 mg/L), Sr (7.97–34.22 mg/L, average 23 mg/L) and B (3.56–13.96 mg/L, average 9.21 mg/L) in the waters of Lake Taloe.

The enrichment of Lake Taloe waters with chemical elements occurs during interaction in the water–rock system under the superimposed influence of the processes of evaporative concentration. The interaction of these processes can be judged by the Gibbs diagrams [54] (Figure 7).

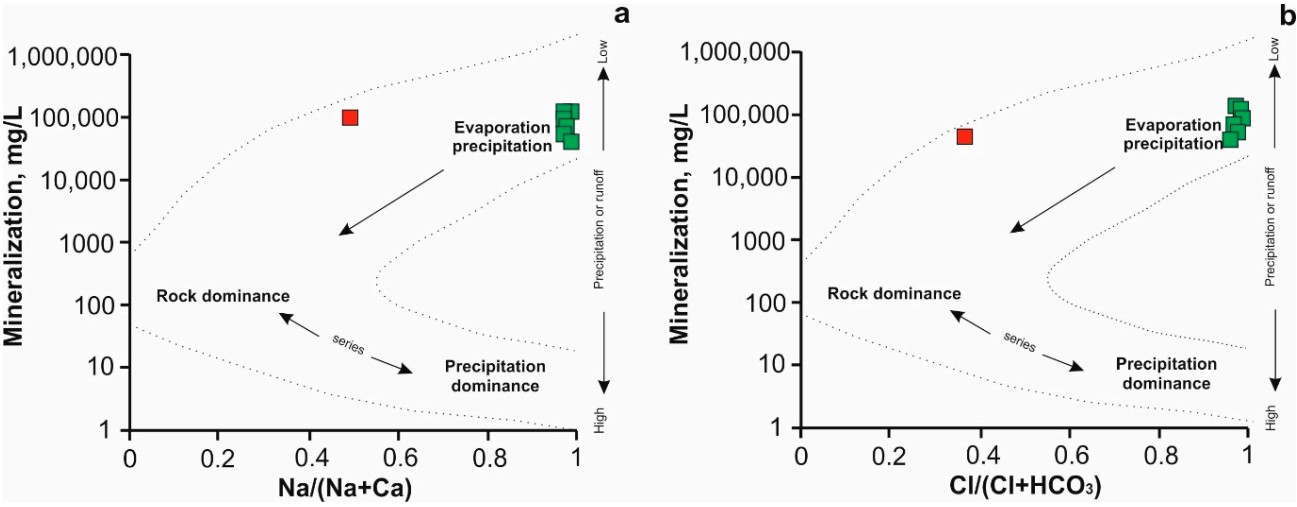

**Figure 7.** Dependence of mineralization on Na/(Na+Ca) (**a**) and Cl/(Cl+HCO3) (**b**). Note. Red points—underground water, green—surface water and pore water of lake sediments.

According to the figure, the position of the points on the diagram indicates the influence of mainly evaporative concentration processes on the formation of the water composition in a semiarid climate. Seasonal changes and humidity and their variations from year to year influence it most of all. A chloride ion was used to illustrate the evaporation process, which accumulates in the solution and is not removed when interacting with rocks, and also does not form secondary minerals at this level of water mineralization [55].

The position of points in the diagram of the dependence of the concentration of the sulfate ion on the concentration of the chloride ion below the line indicates the removal of the sulfate ion from lake waters during secondary mineral formation. This process is confirmed by the species diversity of sulfate minerals in the sediments of Lake Taloe. A similar process is observed for the bicarbonate ion (Figure 8a).

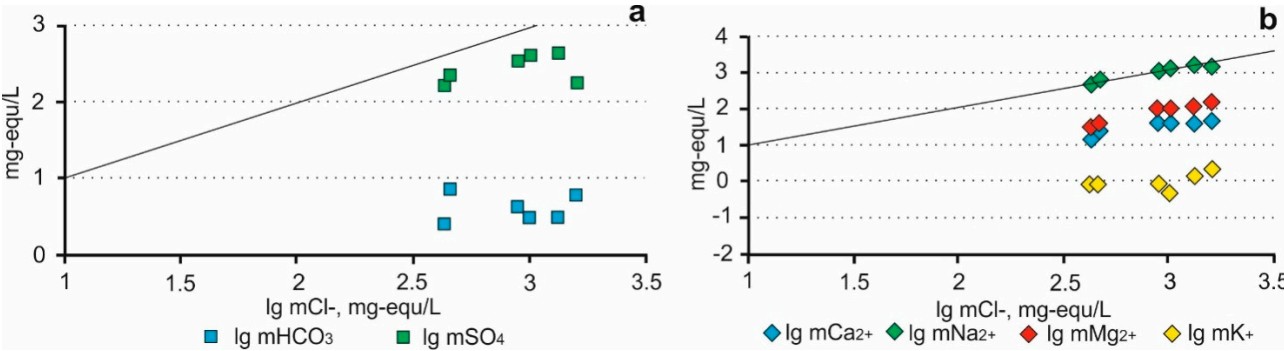

**Figure 8.** Dependence of the concentrations of the main anions (**a**) and cations (**b**) on the concentration of the chloride ion in Lake Taloe waters. A line with an inclination of 45° corresponds to the position of the points expected for simple evaporation.

The position of points in Figure 8b indicates a predominantly proportional concentration of the sodium ions in the process of evaporative concentration along with the chloride ions, which indicates the leading role of this process in the formation of Lake Taloe waters. In the lake under consideration, removing magnesium ions from the solution is observed, which is due to the entry of magnesium into secondary minerals. This process is activated by evaporative concentration. A similar process is observed for calcium ions. The evaporation processes have practically no effect on the concentration of potassium ions in the solution. The position of the points on the graph is due to their low prevalence in the enclosing rocks and the manifestation of secondary mineral formation processes.

Thus, it can be assumed that the formation of the chemical composition of Lake Taloe waters is significantly influenced by the processes of evaporative concentration along with the enrichment of water with chemical elements due to the weathering of the underlying rocks.

The calculation results of the saturation of various types of water to minerals are shown in Table 3. They characterize the possibility of destruction or formation of a particular mineral for each water type.

**Table 3.** The degree of saturation of various water types in Lake Taloe to minerals (disequilibrium indices).

| Mineral | Chemical Formula of The Mineral | UnderGround Waters | Surface Waters | | | Pore Waters of Lake Sediments | | |
|---|---|---|---|---|---|---|---|---|
| | | Xa-60 | 1 | 2 | 3 | 4 | 5 | 6 |
| **Hydroxides** | | | | | | | | |
| Goethite | FeOOH | 3.00 | −0.06 | −0.41 | −1.60 | 0.18 | 0.16 | −0.36 |
| Lepidocrocite | FeOOH | −2.10 | −5.1 | −5.4 | −6.6 | −4.9 | −4.9 | −5.4 |
| Bayerite | Al(OH)$_3$ | 6.50 | 5.10 | 6.50 | 5.20 | 4.90 | 5.40 | 5.10 |
| Gibbsite | Al(OH)$_3$ | 3.90 | 2.40 | 3.80 | 2.60 | 2.30 | 2.70 | 2.50 |
| **Oxides** | | | | | | | | |
| Hematite | Fe$_2$O$_3$ | 11.00 | 4.60 | 3.90 | 1.50 | 5.10 | 5.10 | 4.00 |
| Quartz | SiO$_2$ | 1.40 | 0.05 | −0.01 | 0.49 | 0.14 | 0.06 | 0.14 |
| Quartz (cryptocrystalline variety) | SiO$_2$ | 0.70 | −0.61 | −0.67 | −0.17 | −0.52 | −0.60 | −0.52 |
| Cristobalite | SiO$_2$ | 0.01 | −1.30 | −1.40 | −0.86 | −1.20 | −1.30 | −1.20 |
| Tridymite | SiO$_2$ | 0.40 | −0.90 | −0.97 | −0.47 | −0.82 | −0.89 | −0.81 |
| **Silicates and aluminosilicates** | | | | | | | | |
| Nontronite | Na$_{0.3}$Fe$_2$(Si, Al)$_4$O$_{10}$(OH) × nH$_2$O | 22.00 | 14.00 | 17.00 | 15.00 | 14.00 | 14.00 | 14.00 |
| | Fe$_{0.2924}$Mg$_{2.9}$Al$_{1.6984}$Si$_{3.935}$O$_{10}$(OH)$_2$ | 9.20 | 0.78 | 2.50 | 1.90 | 0.90 | 1.30 | 1.20 |
| | Al$_2$Si$_4$O$_{10}$(OH)$_2$ | 22.00 | 14.00 | 17.00 | 16.00 | 14.00 | 15.00 | 15.00 |
| Montmorillonite group | Ca$_{0.187}$Na$_{0.0205}$K$_{0.0205}$Fe$_{0.141}$Mg$_{0.336}$Al$_{1.59}$Si$_{3.93}$O$_{10}$(OH)$_2$ | 18.00 | 10.00 | 11.00 | 11.00 | 10.00 | 11.00 | 11.00 |
| | K$_{0.3}$Al$_{1.9}$Si$_4$O$_{10}$(OH)$_2$ | 15.00 | 8.00 | 10.00 | 9.80 | 7.90 | 8.60 | 8.60 |
| | Mg$_{0.6525}$Fe$_{0.335}$Al$_{1.47}$Si$_{3.82}$O$_{10}$(OH)$_2$ | 18.00 | 10.00 | 11.00 | 10.00 | 11.00 | 11.00 | 11.00 |
| | Na$_{0.27}$Ca$_{0.1}$K$_{0.02}$Fe$_{0.19}$Mg$_{0.22}$Al$_{1.58}$Si$_{3.94}$O$_{10}$(OH)$_2$ | 18.00 | 11.00 | 12.00 | 12.00 | 11.00 | 12.00 | 12.00 |
| | Ca$_{0.15}$Al$_{1.9}$Si$_4$O$_{10}$(OH)$_2$ | 23.00 | 16.00 | 18.00 | 17.00 | 16.00 | 16.00 | 16.00 |
| | Mg$_{0.165}$Al$_{2.33}$Si$_{3.67}$O$_{10}$(OH)$_2$ | 22.00 | 14.00 | 17.00 | 16.00 | 14.00 | 15.00 | 15.00 |
| Beidellite | Ca$_{0.165}$Al$_{2.33}$Si$_{3.67}$O$_{10}$(OH)$_2$ | 22.00 | 14.00 | 17.00 | 16.00 | 14.00 | 15.00 | 15.00 |
| | Na$_{0.33}$Al$_{2.33}$Si$_{3.67}$O$_{10}$(OH)$_2$ | 24.00 | 17.00 | 20.00 | 19.00 | 18.00 | 18.00 | 18.00 |
| Chlorite group | Mg$_6$Si$_4$O$_{10}$(OH)$_8$ | −26.00 | −30.0 | −40.0 | −41.00 | −30.0 | −34.0 | −30.0 |
| Illite | K$_{0.63}$Na$_{0.017}$Ca$_{0.145}$Fe$_{0.327}$Mg$_{0.349}$Al$_{1.9}$Si$_{3.421}$O$_{10}$(OH)$_2$ | 22.00 | 15.00 | 16.00 | 15.00 | 15.00 | 16.00 | 16.00 |
| K-Mg-illite | K$_{0.6}$Mg$_{0.25}$Al$_{2.3}$Si$_{3.5}$O$_{10}$(OH)$_2$ | 25.00 | 18.00 | 21.00 | 19.00 | 18.00 | 19.00 | 19.00 |
| Illite | K$_{0.5}$Al$_{2.5}$Si$_{3.5}$O$_{10}$(OH)$_2$ | 27.00 | 20.00 | 23.00 | 21.00 | 20.00 | 21.00 | 21.00 |
| Anorthite | CaAl$_2$Si$_2$O$_8$ | −2.70 | −7.30 | −6.60 | −8.20 | −7.50 | −7.30 | −7.40 |
| Albite | NaAlSi$_3$O$_8$ | −0.58 | −0.04 | −0.18 | −0.14 | 0.21 | 0.56 | −3.77 |
| Potassic feldspar | KAlSi$_3$O$_8$ | 0.15 | −0.52 | −0.34 | −0.31 | −0.59 | 0.09 | −3.98 |
| **Carbonate** | | | | | | | | |
| Magnesite | MgCO$_3$ | −0.65 | −1.30 | −2.70 | −2.00 | −1.80 | −2.10 | −0.80 |
| Lansfordite | MgCO$_3$ × 5H$_2$O | −2.80 | −3.70 | −5.00 | −4.30 | −4.20 | −4.60 | −3.40 |
| Witherite | BaCO$_3$ | −8.30 | −10.0 | −10.0 | −11.0 | −11.0 | −12.0 | −12.0 |
| Aragonite | CaCO$_3$ | 2.50 | 2.20 | 0.57 | 1.70 | 1.80 | 1.50 | 2.30 |
| Calcite | CaCO$_3$ | 2.80 | 2.60 | 0.92 | 2.00 | 2.20 | 1.80 | 2.60 |
| Dolomite | CaMg(CO$_3$)$_2$ | 3.90 | 3.00 | −0.10 | 1.70 | 2.10 | 1.40 | 3.60 |
| Siderite | FeCO$_3$ | −3.00 | −7.60 | −7.10 | −6.80 | −7.80 | −7.30 | −6.90 |
| Rhodochrosite | MnCO$_3$ | −2.40 | −5.20 | −5.60 | −1.20 | −6.20 | −7.00 | −1.90 |
| Strontianite | SrCO$_3$ | 0.38 | −0.30 | −1.80 | −1.40 | −0.80 | −1.10 | −0.71 |
| **Sulfates** | | | | | | | | |
| Epsomite | MgSO$_4$ × 7H$_2$O | −7.20 | −5.80 | −6.20 | −6.20 | −6.00 | −6.10 | −6.20 |
| Kieserite | MgSO$_4$ × H$_2$O | −64.00 | −63.00 | −63.00 | −63.00 | −63.00 | −63.00 | −63.00 |
| Thénardite | Na$_2$SO$_4$ | −61.00 | −50.00 | −52.00 | −51.00 | −50.00 | −50.00 | −50.00 |
| Baryte | BaSO$_4$ | 0.14 | 0.31 | 1.40 | −0.29 | 0.05 | −0.20 | −1.40 |
| Anhydrite | CaSO$_4$ | −3.50 | −1.40 | −2.20 | −1.70 | −1.40 | −1.50 | −1.90 |
| Gypsum | CaSO$_4$ × 2H$_2$O | −0.97 | 0.32 | −0.28 | 0.00 | 0.31 | 0.31 | 0.07 |
| Melanterite | FeSO$_4$ × 7H$_2$O | −7.80 | −10.00 | −8.80 | −9.20 | −10.00 | −9.50 | −11.0 |
| Jarosite | KFe$_3$(SO$_4$)$_2$(OH)$_6$ | −33.00 | −37.00 | −36.00 | −39.00 | −37.00 | −35.00 | −38.0 |
| Celestine | SrSO$_4$ | 0.47 | 2.10 | 1.40 | 1.20 | 2.00 | 1.90 | 1.10 |
| **Haloids** | | | | | | | | |
| Halite | NaCl | −15.00 | −4.70 | −5.90 | −5.80 | −4.40 | −3.90 | −3.70 |
| Hydrohalite | NaCl × 2H$_2$O | −14.00 | −4.50 | −5.70 | −5.50 | −4.20 | −3.80 | −3.50 |
| Antarcticite | CaCl$_2$ × 6H$_2$O | −95.00 | −84.00 | −86.00 | −86.00 | −84.0 | −84.0 | −83.00 |
| Bischofite | MgCl$_2$ × 6H$_2$O | −100.00 | −93.00 | −95.00 | −95.00 | −93.0 | −93.0 | −91.00 |
| Sylvite | KCl | −17.00 | −10.00 | −11.00 | −10.00 | −10.0 | −9.10 | −8.40 |
| Scacchite | MnCl$_2$ | −52.00 | −43.00 | −44.00 | −40.00 | −43.0 | −43.0 | −38.00 |

The studied lake waters are undersaturated with respect to a number of primary aluminosilicates (anorthite); haloids (halite, hydrohalite, antarcticite, bischofite, sylvite, scacchite); sulfates (epsomite, kieserite, thénardite, anhydrite, melanterite, jarosite); secondary aluminosilicate minerals (chlorite group) and carbonates (magnesite, lansfordite, witherite, siderite, rhodochrosite, strontianite); some oxides and hydroxides (lepidocrocite, cristobalite, tridymite, and quartz (cryptocrystalline variety)).

Many of these minerals (anorthite, minerals of the chlorite group, etc.) are rock-forming minerals of the migration environment, and accordingly, serve as sources of enrichment for underground waters with elements throughout the entire movement from the feeding areas to the discharge areas, their presence in sediments is unlikely.

Positive non-uniformity indices indicate the equilibrium of groundwater with sulfates (baryte, celestine), carbonates (strontianite, aragonite, calcite, dolomite), oxides (quartz, cristobalite, tridymite, chalcedony), aluminosilicates (nontronite, montmorillonite with various predominant cations, beidellite, chlorite, illite, and potassic feldspar).

According to the values of the disequilibrium indices, the waters of the lake, in comparison with underground waters, are also more balanced with some carbonates (aragonite, calcite, and dolomite), sulfates (gypsum), aluminosilicates (montmorillonite, beidellite, illite). Additionally, the pore waters of lake sediments also contain goethite and quartz.

It is worth noting that halides are not in equilibrium with any type of water under study, which allows for assuming that their formation is only in the process of evaporative concentration.

All the mentioned minerals can create new formations from the waters and remove the corresponding elements from them. The physical and chemical calculations performed are in agreement with mineralogical studies and confirm the hydrogenic mechanism of sedimentation. Accordingly, in the composition of mineral sediments, the presence of these minerals is observed in quantities and ratios that reflect the specific chemical composition of the aquatic environment from which they are formed.

The obtained physico-chemical calculations can be used as a basis for further studies of the processes of lake formation and their sediments.

*4.3. Mineral Associations and Patterns of Mineral Accumulation*

Based on the obtained set of mineral phases, two genetic types of associations can be distinguished: terrigenous and hydrogenic. The terrigenous association includes minerals that are not soluble in water: clay minerals, quartz, feldspar, analcime. The hydrogenic association includes mineral species formed by evaporative concentration at the expense of underground and surface waters—minerals of the halides, sulfates, and carbonates groups.

Mineralogically, the terrigenous association coincides with the composition of the underlying rocks. The terrigenous association includes minerals: quartz, feldspars, zeolite and clay minerals, which were formed due to the destruction of the underlying rocks. Clay minerals are epigenetic minerals formed in lake conditions due to the interaction of groundwater with feldspar. It is generally believed that clay minerals can be formed by the destruction of plagioclase and potassic feldspar, resulting in montmorillonite, illite, and minerals of the chlorite group formation.

The mineral set of the hydrogenic association is mainly represented by water-soluble carbonate, sulfate minerals (with different solubility degrees), and halides. The carbonate minerals are dominated by calcite, distributed over the entire area and depth of the lake. It is worth noting that calcite is distributed relatively evenly, and there are no direct dependencies on the depth and laterality of lakes, as in the Meighan sediments (Iran) [14]. In Lake Taloe sediments, the uniform distribution of calcium carbonate against the background of constantly low dolomite contents (which is very unstable under the influence of weathering and transportation processes [14]) suggests that some of the calcite is of a clastic origin.

During this study, the vertical zoning of the qualitative change in mineral species was recorded. Thus, in the upper part of the section (0–20 cm from the surface), the most

water-soluble minerals (halides and sulfates) are recorded, which in dry periods form a salt crust with a thickness of 2–3 cm. Moreover, at the level of 0–10 cm from the surface, their concentrations are much higher. In the range of 20–50 cm in the humus horizon, the mineral composition is relatively homogeneous and is represented by terrigenous, carbonate, and clay minerals; water-soluble minerals are less common. Within this interval, small layers of gleic sediments are observed. At a depth of 50–90 cm, the mineral composition is similar to the interval of 10–50 cm, there are intensively gleic (blue-gray) areas. At a depth of 90–100 cm, there is a typical salt marsh gleic bedrock with a characteristic smell of hydrogen sulfide (Figure 2).

Lateral zoning is reflected both in the quantitative and qualitative mineral ratio. Carbonate minerals are distributed throughout the shores of the lake in significant amounts (10–29% of the total precipitation). There is an increase in the amount of gypsum and halite in the direction from the shoreline of the lake to the central one, up to 13% and 14%, respectively. It is worth noting that there is no gypsum directly near the coastal zone.

The precipitation of hydrogenic minerals from the waters of intermittent lakes is a complex process regulated both by the humidity and by the processes of evaporative concentration. According to the scheme for brine evolution (according to [56]), the precipitation of hydrogenic minerals begins, as a rule, with calcium carbonate—calcite. Due to its precipitation, the lake water becomes more concentrated, and thus it becomes necessary to consider the influence of the magnesium ion, which contributes to dolomite formation. Dolomite formation is possible only when the Mg/Ca ratio in the solution increases, which is impossible without the precipitation of sulfate minerals. Due to the continuous action of water with rock-forming minerals, as a result of the hydrolysis of aluminosilicates, $Na^+$, $Ca^{2+}$, and $Mg^{2+}$ ions enter the water, which leads to the formation of sulfate salts. According to the scheme [14], sulfates fall mainly in the gypsum phase [13], but the sulfates blödite, wattevillite, epsomite, thénardite, eugsterite, baryte, and celestine were also identified in the sediments of Lake Taloe. It is worth noting that gypsum in Lake Taloe sediments is distributed mainly in the upper parts of the sections (0–20 cm from the surface) and the central part of the lake. This is due to the process of its formation in lake sediments, namely, in the process of oxidation of hydrogen sulfide located in the coastal part of the atmospheric air, under the influence of ground-level ozone, free sulfurous acids (Equation (1)), and then sulfuric acid (Equation (2)) are formed.

$$H_2S + O_3 \rightarrow H_2SO_3 \tag{1}$$

$$2H_2SO_3 + O_2 \rightarrow 2H_2SO_4 \tag{2}$$

Under high humidity conditions, these acids react with calcium-containing rocks to form calcium sulfate (Equation (3)).

$$H_2SO_4 + CaCO_3 + H_2O \rightarrow CaSO_4 + H_2O + CO_2 \tag{3}$$

Additionally, the deposition of gypsum in the upper parts of the sediments (0–20 cm) is facilitated by its low solubility. When reducing the $Ca^{2+}$ and $Mg^{2+}$ ions, the $Na^+$ content in the brine increases, resulting in the sodium sulfate–thénardite precipitation. In the sediments, the content of thénardite is observed in negligible amounts; this is due to its high solubility. Further, there is an increase in the content of chloride ions in the solution, and the precipitation of halite begins. In general, the order of precipitation of cations and anions can be represented as follows: $Na^+ > Mg^{2+} > Ca^{2+}$ and $Cl^- > SO_4^{2-} > HCO_3^- > CO_3^{2-}$.

According to the scheme presented in the work by [14], halite is distributed only in the evaporation zone (mainly in the central zone of the lake), while halite in Lake Taloe is distributed throughout the entire lateral part of the lake, unevenly from the coastal part to the central one. In the authors opinion, this is due to its high solubility, and the presence of microfractures in the lake sediments through which the halite dissolved in the waters is transported and partially deposited in the sediments. This process will



continue until halite precipitation begins and $Mg^{2+}$ becomes the predominant cation in the solution, contributing to the formation of sulfates. Similar chemical transformations are also observed in intermittent salt lakes around the world: Lake Sambhar [17], Lake Seife [7], Lake Meighan [14]. The work of L. Abdi provides a scheme for modeling the chemical and mineralogical distribution, which partially corresponds to the patterns of distribution of mineral phases in Lake Taloe. The lake sediments are characterized by the presence of a wider set of sulfate minerals. Figure 9 shows an updated scheme of mineral distribution in Lake Taloe.

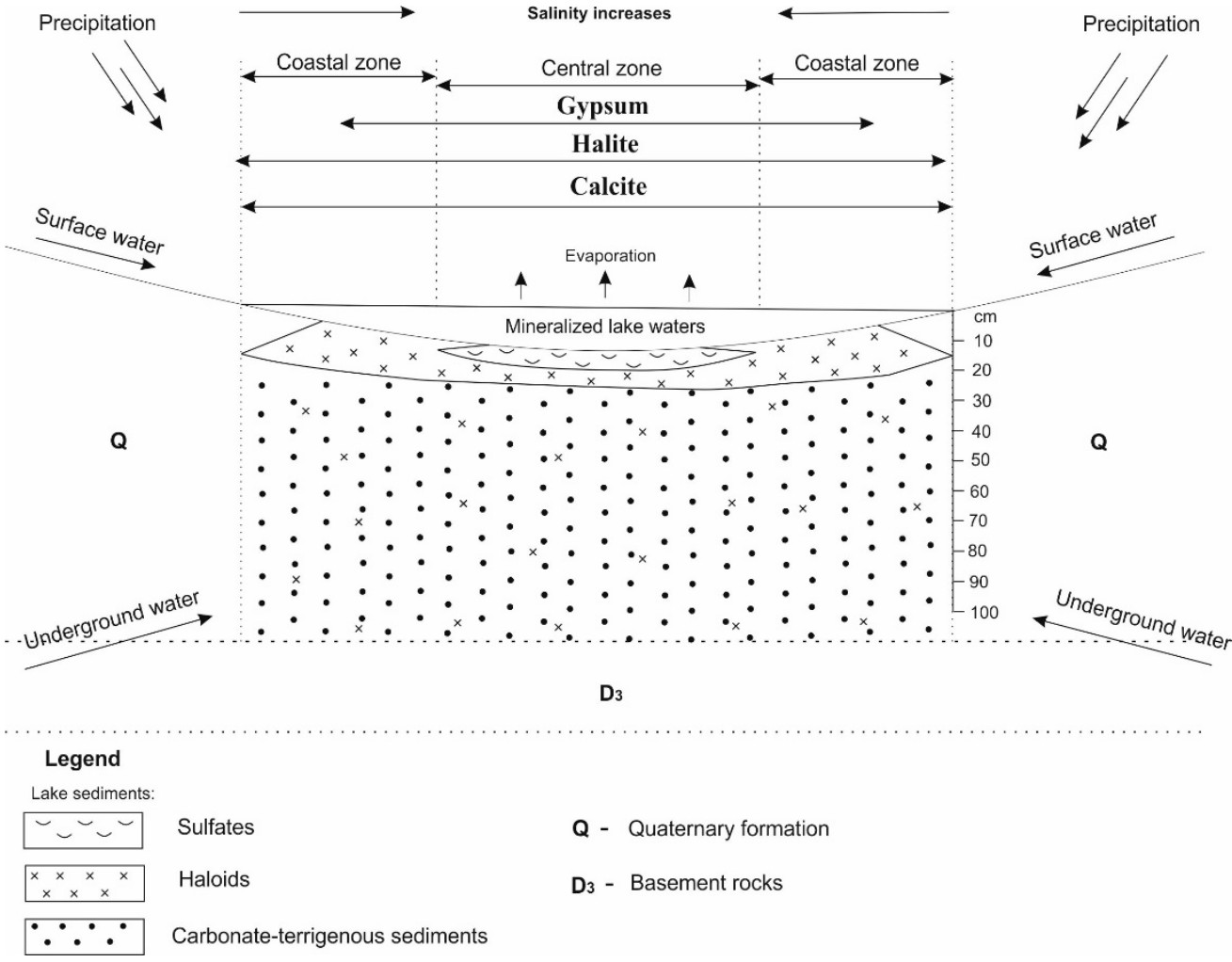

**Figure 9.** Model-based diagram of the mineralogical distribution of Lake Taloe sediments [14].

The presented diagram of mineralogical distribution can be used for various intermittent lakes but adjusted for the region's individual physical, chemical, and climatic conditions.

## 5. Conclusions

Based on the data obtained, the following conclusions can be drawn:

(1) Instrumental methods were used to identify minerals in lake sediments of Lake Taloe: oxides—quartz, hematite; sulfides—pyrite; slumosilicates—minerals of the chlorite group, illite, mixed-layer formations of illite—montmorillonite series, albite, potassic feldspar; carbonates—calcite, dolomite, aragonite; sulfates—gypsum, thénardite, blödite, wattevillite, epsomite, eugsterite, celestine, baryte; halides—halite.

(2) The resulting mineral species, according to genetic characteristics, are divided into two associations: terrigenous and hydrogenic. The terrigenous association combines minerals that are not soluble in water and are formed as a result of the bedrock fracture. The hydrogenic association combines typical water-soluble minerals formed due to the evaporative concentration of water and the effect of groundwater on sediments.

(3) Vertical and lateral zoning of precipitation distribution is recorded. Vertical zoning is reflected in the qualitative ratio of mineral species: sulfates and halides are concentrated mainly in the upper part of the section. Lateral zoning consists in an increase in the content of gypsum and halite from the periphery toward the center of the lake.

(4) The procedure for the deposition of minerals from lake waters is established. It is expressed in the following form: for cations—$Na^+ > Mg^{2+} > Ca^{2+}$, for anions $Cl^- > SO_4^{2-} > HCO_3^- > CO_3^{2-}$.

(5) Physical and chemical calculations demonstrate that the sulfate–chloride waters of the lake are in equilibrium with hydroxides, oxides, aluminosilicates, carbonates, and sulfates. The formation of minerals is significantly influenced by evaporative concentration together with the processes of bedrock weathering.

Thus, the results obtained can serve as a basis for further physical and chemical modeling of the processes occurring in salty intermittent lakes of a semiarid climate to understand the conditions for forming such features.

**Author Contributions:** Conceptualization, M.O.K., E.M.D. and P.A.T.; Data curation, A.I.C.; Formal analysis, M.O.K.; Funding acquisition, M.O.K. and P.A.T.; Methodology, M.O.K., E.M.D. and P.A.T.; Project administration, E.M.D., P.A.T. and A.I.C.; Resources, M.O.K.; Software, M.O.K., E.M.D., A.L.A. and A.N.N.; Validation, P.A.T. and A.I.C.; Visualization, A.L.A.; Writing—original draft, M.O.K., E.M.D. and A.N.N.; Writing—review and editing, M.O.K., E.M.D., P.A.T. and A.I.C. All authors have read and agreed to the published version of the manuscript.

**Funding:** The reporting study was funded by the Russian Foundation for Basic Research, project number 19-35-90056, grant number 0721-2020-0041 from the Ministry of Education and Science of Russia, as well as by TPU development program.

**Data Availability Statement:** Data confirming the results obtained can be obtained upon request from the first author.

**Acknowledgments:** We are grateful to Tatyana S. Nebera for helping with the interpretation of XRD analysis results and Evgeniy V. Korbovyak for carrying out the SEM analysis.

**Conflicts of Interest:** The authors declare no conflict of interest.

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
