# Peer review of "Taloe—Sedimentation in an Intermittent Lake (Russian Federation, Republic of Khakassia)"

_minerals, doi:10.3390/min11050522_

Round 1

Reviewer 1 Report

The text and conclusion requires the major revision. The formation of the chemical and mineralogical composition of salt lakes is a multifactor process. Together with mineral sedimentation as a result of the evaporation, the processes of dissolving of minerals and forming of new minerals as well as bacterial reduction of sulfates and oxidation of organic components and sulfides can be registered. Therefore in the lake under consideration, there are the products of all these processes, namely soda, carbonates, chlorides, and sulfates. It is important that different geochemical processes have an important role in the formation of different stages of the lake. For example, if the concentration of sulfates in water is increasing, pH of lake water is decreasing and there is not equilibrium with gypsum sedimentation and other sulfate minerals, the sulfate type of lake is forming. The chloride lake forms in conditions of high evaporation and the process of sulfate-reduction. The results obtained by the authors give the possibility to consider these processes and stages of salt lake formation in more detail and support a more correct model of sedimentation in the Taloe lake.        

Author Response

We thank the reviewer for his careful reading of the article. We have addressed all your comments in the revised manuscript. Please see the attached file for more detail. Corrections in the text of the article are highlighted in green.

Comments on the comments of the reviewer

  1. Introduction

This article devoted to study the nature of mineral species formation in sulfate-chloride intermittent lakes in a semiarid climate and calculate the physicochemical equilibria of the watermineral system to confirm the experimental results. But just one lake was considered. In the article

the waters and sediments in the Taloe lake, located in the steppe zone with a predominantly semiarid climate were exanimated. In this case it is important to consider the models of sedimentation in sulfate-chloride intermittent lakes of other arid and semiarid regions like Chine, Mongolia, Baikal region etc. for comparison with this type of sedimentation.

Corrections on the line33-37 and 42-43. Information on lake types added.

  1. Characteristics of the area and object of research

According to the publication [28] the age of the loose sediments is estimated as Eopleistocene – Lower Neo-Pleistocene and the rate of sedimentation is estimated at about 2 mm per year. The thickness of loose deposits varies from 0.5 to 2.5 m, reaching in the lower parts of the terrain 5– 10 m. In this case, the calculation of the age of lake deposits consists of no longer than 5 000 years. What method was used to determine the age of lake sediments? The features of mineralogical association (e.g. aragonite) is evidence that the process of lake sedimentation could be not later than Holocene also.

It was assumed that unconsolidated sediments of northwest part of South-Minusinsk basin (where Taloe lake is located) formed in Eopleistocene and early Neo-Pleistocene. Lake sediments themselves formed in Holocene. Considering that average sedimentation rate is about 2 mm per year (according to Rogozin [Rogozin, D.Yu. Meromictic lakes of the North Minusinsk depression: patterns of stratification and ecology of phototrophic sulfur bacteria. Krasnoyarsk: Publishing House: IF SB RAS. 2018, 241 (in Russian)]), the studied cross section age is about 500 years. Corrections on the line 64-65, 70-71, 73-74 and 96-100.

  1. Materials and methods

3.1. Sampling

It is not clear again what age of sampling sediments. All cores had a depth of 1 meter from the surface. What is the age of these deposits in the coastal part and in the littoral part? It is important for the model of sedimentation under study.

Sediments from shore and central parts of the lake formed in Holocene. Corrections on the line 96-97.

  1. Results and discussion

4.1. Oryctognosy of lake sediments

The term “Oryctognosy” was applied in 18-19 centuries and it is better to be replaced with the modern term “Mineralogy’.

The term was changed. Corrections on the line 178.

4.1.1. Oxides, sulfides, and aluminosilicates

Line 190: “Quartz is considered the main terrigenous mineral, while its partial hydrogenic origin is not excluded”. Quartz is a mineral type and can be precipitated from hydrothermal water. In lake water it is possible just in the form of silica gel and diatoms scelets but not in form of quartz.

Corrections on the line 196-198. It wasn’t meant to say that quartz itself precipitates from lake water, only that there is possibility of hydrogenous silica (including silica gel) precipitation.

Line 197: “Pyrite is distributed throughout the entire area of the lake. The presence of hydrogen sulfide in the lake creates a favorable environment for the formation of iron sulfide”. Explain these sentences. What is a favorable environment? What is the difference between conditions of pyrite and gypsum formation in arid and semiarid conditions.? What is the role of sulfate-reducing bacteria?

Sulfate reduction is typical for salt lakes. As a rule, there are sulfate-reducing bacteria, and one of products of sulfate reduction is hydrogen sulfate. Strong hydrogen sulfate smell was noticed during the field work. This process is described in detail in works of Borzenko S. V [Borzenko, S.V. The main conditions for the formation of the chemical composition of the waters of saline and brackish lakes in Eastern Transbaikalia. Geochemistry. 2020, 65, 12, 1212-1230 (in Russian)]. Unlike gypsum, pyrite formation necessarily requires reductive conditions.

Line 213: “however, the literature notes the possibility of the formation of hydrogenated sodium feldspar in salt lakes [50]”. It should be specified the water conditions (pH, mineralization), which influence the albite precipitation. It is worth noting that there are cations and anions in the solutions but not minerals.

Corrections on the line 223-224. It is generally accepted that albite in lake sediments is terrigenous. But it is noted in Borzenko’s works [Borzenko, S.V. The main conditions for the formation of the chemical composition of the waters of saline and brackish lakes in Eastern Transbaikalia. Geochemistry. 2020, 65, 12, 1212-1230 (in Russian)] that there is hydrogenous albite catched by sediment traps. Never the less, its part is negligibly small in comparison to terrigenous albite. According to physicochemical calculations, formation of hydrogenous albite at Taloe lake is possible at relatively high pH (> 7.5) and mineralization (> 85 g/l).

4.2. Geochemical features and mineral-forming capacity of underground and lake waters

Line 324. “According to the figure, the position of the points on the diagram indicates the influence of mainly evaporative concentration processes on the formation of the water composition in a semiarid climate”. The process of mineral precipitation/evaporation in the lakes depends on climatic factors (e.g. temperature, humidity, seasonality). How did these factors affect the lake mineralogy?

Corrections on the line 337-338. At the present time season and humidity are what influences it most. Formation of various minerals including halides is possible during the «dry» season. Carbonates and sulfates form while halides dissolve when humidity increases.

Line 361. Cristobalite, tridymite are the high-temperature polymorphic modifications of quartz. These minerals cannot be presented in the water solution of the lake just in deposits as terrigenous components.

As a result of physicochemical calculations, it is assumed that cristobalite and tridymite appear as intermediate phases during the recrystallization of silica in silica – «opal-cristobalite-tridymite» – quartz row. It’s important to notice that these phases are unstable. According to physicochemical calculations, cristobalite and tridymite are non-equilibrium in surface lake waters.

Line 382. “Accordingly, in the composition of mineral sediments, the presence of these minerals is observed in quantities and ratios that reflect the specific chemical composition of the aquatic environment from which they are formed”. Clarify this sentence. Such minerals as quartz, feldspar, chlorite do not precipitate from water solution in lakes.

This is concluded from thermodynamic calculations, which agree with similar studies made by other researchers.

Line 408. “Thus, in the upper part of the section (0–20 cm from the surface), the most watersoluble minerals (halides and sulfates) are recorded, which in dry periods form a salt crust with a thickness of 2–3 cm”. For which time this crust was formed? What is the time of formation of all sections?

How do the concentrations of different minerals vary in the layers of the different cross-sections?

2-3 cm thick salt crust forms in conditions of high solar activity and no precipitation in 24-48 hours (while most of lake water evaporates). Precipitation leads to its dissolution. Unfortunately, our sampling scheme didn’t allow doing a detailed study of smaller layers.

  1. Conclusion

Line 487. “Lateral zoning is reflected in the quantitative ratio of minerals: from the shoreline to the center of the reservoir, gypsum, and halite content increases”. How did the lake mineralization change in different periods of sedimentation in the Eopleistocene – Lower Neo-Pleistocene? What did factors influence the mineral formation? The hydrological regime, climatic changes, seasonality?

The formation of the chemical and mineralogical composition of salt lakes is a multifactor process. Together with mineral sedimentation as a result of the evaporation, the processes of dissolving of minerals and forming of new minerals as well as bacterial reduction of sulfates and oxidation of organic components and sulfides can be registered. Therefore in the lake under consideration, there are the products of all these processes, namely soda, carbonates, chlorides, and sulfates. It is important that different geochemical processes have an important role in the formation of different stages of the lake. For example, if the concentration of sulfates in water is increasing, pH of lake water is decreasing and there is not equilibrium with gypsum sedimentation and other sulfate minerals, the sulfate type of lake is forming. The chloride lake forms in conditions of high evaporation and the process of sulfate-reduction. The results obtained by the authors give the possibility to consider these processes and stages of salt lake formation in more detail and support a more correct model of sedimentation in the Taloe lake.

Taloe lake sediments formed in Holocene. Unfortunately, our sampling scheme (discrete sampling at every 10 cm) didn’t allow doing a more detailed study of vertical cross section. Lateral zoning is a result of progressive drying from shore to center. According to conventional scheme of mineral formation in salt lakes (carbonates-sulfates-halides), carbonates sediment first near the shore, then (with continuous drying and closer to center) sulfates do, and halides form at the very end. Without a doubt, climate, seasonal changes and hydrological regime influence the formation and distribution of minerals.

Reviewer 2 Report

General comments:

            This paper presents descriptive results for the dilution, concentration, and mineralization processes at play in a salty, intermittent lake in a semi-arid climate.  Since intermittent lakes exist in a variety of climate and geological settings at present and are present in the geological record world-wide, this work provides useful additional information about these interesting structures.

            A brief summary of how this study site fits into our overall understanding of intermittent lakes, perhaps including named examples of such features from different geological/climatological settings would be very useful to readers who are interested in the overall results, but not the details of the technical procedures.

            As listed below, there are numerous corrections needed for the text.

Specific language issues:

  • When referring to peaks on an XRD spectrum, the correct term is “reflections” not “reflexes”. Replace all occurrences of “reflexes”.
  • The word “subsidence” generally refers to the sinking of landforms. In this paper, I think you mean “settling out” or “precipitation” of solids out of a liquid.  In a beaker of turbid water, for example, the particles settle to the bottom leaving a transparent liquid at the top…  Replace all occurrences of “subsidence”.
  • The word “understudy” refers to a back-up performer, e.g. in an opera. I think you mean “under study” in the sense of “being examined”.
  • Line 28: “used to control the evaporation rate…” reword, this is awkward.
  • Line 72: “map of the actual material…” Figure 1 should probably be top-bottom reversed so it goes from globe to geological map to satellite image.  I don’t understand which map shows “actual material”.  The photographic image shows the actual site.
  • Line 100: “of the section under…” I think this should be “of the area under…”  Figure 2 show a “section”.
  • Line 146: I think it should read “authors”. If not, then which author carried out the calculations?
  • Line 203: “the fracture of sodium plagioclase” should be “the alteration of sodium plagioclase”.
  • Line 204: the comma should be a semicolon, i.e. “feldspar, to a lesser…” should be “feldspar; to a lesser…”
  • Line 227: I have no idea what you mean by “inter-batch water adsorption”.
  • Line 339: The first sentence of this paragraph is awkward. I suggest “concentration of sodium ions…”  The phrase “and the chloride ion” just seems to be hanging out there.
  • Line 350: Again, replace the comma after with a semicolon, “(anorthite);”  also “staccite);” “…jarosite);” also “strontianite); and…”
  • Line 362: What is a “rock-forming migration environment” are there words missing?
  • Line 387: Should “combines” be “includes”?
  • Line 393: What does the sentence beginning with “Quartz,…” mean?
  • Line 405: Do you have a reference for the assertion that dolomite is “very unstable under the influence of weathering and transportation processes…”?
  • Line 419: “late laterals…”?? Do you mean “shores of the lake”?
  • Line 424: What is the “balance of relative humidity…”?
  • Line 457: “author’s” should be “authors’”
  • Line 470: “This diagram of mineralogical…” might be better. What does the word “objects” refer to?
  • Line 488: A bit unclear. Are you saying that the gypsum and halite content increases from the periphery toward the center of the reservoir?
  • Line 488: Any place in the manuscript that you use the term “reservoir”, change it to “lake”. If the lake is actually sometimes used as a reservoir, ignore this comment.
  • Line 499: “objects” is probably the wrong word. How about “features”?

Mineral Nomenclature (Table 1, Table 3 and elsewhere)

  • Chlorite is a mineral group, not a species—a footnote about this would be good.
  • Thenardite should be Thénardite.
  • Astrakhonite should be Blödite.
  • Gibbsit should be Gibbsite.
  • Chalcedony is a cryptocrystalline variety of Quartz, not a separate mineral species
  • Nontronite has a nonstandard chemical formula listed. I think it should be Na0.3Fe3+2(Si,Al)4O10(OH)2.nH2O.
  • The seven Montmorillonite listings in table 3 are confusing. Fe-Mg Montmorillonite is not a mineral species, you probably mean a ferroan, magnesian Montmorillonite (or a ferrian, magnesian montmorillonite—I can’t tell). Why are there two listings for just  Montmorillonite.  All the other listings are chemical variants and not recognized minerals.  Maybe a footnote explaining what you are trying to do would be useful.
  • The three chemical variants of Beidellite are also confusing. There is only one recognized mineral species, Beidellite. The rest are magnesian Beidellite, calcian Beidellite, and sodic Beidellite.
  • Illite is a commonly used geological term, but there is no IMA-approved mineral species, Illite. Also, how can K-illite have less potassium in it than plain old Illite?
  • Why use Anorthite for the calcium end-member of the plagioclase feldspar series, but not Albite for the sodium end-member?
  • Melanterite is a ferrous sulfate, whereas Jarosite is a ferric sulfate. The formulas should indicate this.
  • I don’t believe Staccite is a valid mineral species. A reference for this term is needed.

Is it possible to left-justify the list of minerals in table 3?  The variable indentations from having the text centered imply significance to the apparent indentation, but there isn’t any!

Technical Issues

  • Scattered throughout the manuscript sodium ions are listed as divalent, i.e. Na2+. This is wrong. Sodium is monovalent, i.e. Na+.  These must be corrected.
  • Lines 442-446: I think the equations shown should be balanced. I don’t know what O3 is. In equation 2, there are five oxygen atoms on the left, but eight oxygen atoms on the right…?
  • In the paragraph starting at Line 288, I think the type-setting of HCO3- is garbled. The carbonic acid sequence in question is H2CO3, HCO3-, CO32-.

Author Response

We thank the reviewer for his careful reading of the article. We have addressed all your comments in the revised manuscript. Please see the attached file for more detail. We would like to express our special gratitude for the help in correcting language errors. Corrections in the text of the article are highlighted in red.

Comments on the comments of the reviewer

Specific language issues:

  • When referring to peaks on an XRD spectrum, the correct term is “reflections” not “reflexes”. Replace all occurrences of “reflexes”.

Throughout the text, the term "reflexes" was replaced by "reflections".

  • The word “subsidence” generally refers to the sinking of landforms. In this paper, I think you mean “settling out” or “precipitation” of solids out of a liquid.  In a beaker of turbid water, for example, the particles settle to the bottom leaving a transparent liquid at the top…  Replace all occurrences of “subsidence”.

Throughout the text, the term "subsidence" was replaced by "precipitation".

  • The word “understudy” refers to a back-up performer, e.g. in an opera. I think you mean “under study” in the sense of “being examined”.

Throughout the text, the term "understudy" was replaced by "under study".

  • Line 28: “used to control the evaporation rate…” reword, this is awkward.

Corrected the phrase "used to control the evaporation rate" to "used to assess the evaporation rate".

  • Line 72: “map of the actual material…” Figure 1 should probably be top-bottom reversed so it goes from globe to geological map to satellite image.  I don’t understand which map shows “actual material”.  The photographic image shows the actual site.

Figure 1 has been fixed. Footnotes have been added to the figure caption.

  • Line 100: “of the section under…” I think this should be “of the area under…”  Figure 2 show a “section”.

We made a mistake with the figure number. Corrected.

  • Line 146: I think it should read “authors”. If not, then which author carried out the calculations?

Corrected. Replaced "author" with "authors".

  • Line 203: “the fracture of sodium plagioclase” should be “the alteration of sodium plagioclase”.

The phrase was corrected.

  • Line 204: the comma should be a semicolon, i.e. “feldspar, to a lesser…” should be “feldspar; to a lesser…”

Corrected.

  • Line 227: I have no idea what you mean by “inter-batch water adsorption”.

The phrase was deleted. This change did not change the meaning of the sentence.

  • Line 339: The first sentence of this paragraph is awkward. I suggest “concentration of sodium ions…”  The phrase “and the chloride ion” just seems to be hanging out there.

The proposal was corrected.

  • Line 350: Again, replace the comma after with a semicolon, “(anorthite);”  also “staccite);” “…jarosite);” also “strontianite); and…”

Corrected.

  • Line 362: What is a “rock-forming migration environment” are there words missing?

The proposal was corrected.

  • Line 387: Should “combines” be “includes”?

Replaced "combines" with "includes".

  • Line 393: What does the sentence beginning with “Quartz,…” mean?

Rewrote the proposal.

  • Line 405: Do you have a reference for the assertion that dolomite is “very unstable under the influence of weathering and transportation processes…”?

Reference added.

  • Line 419: “late laterals…”?? Do you mean “shores of the lake”?

Yes, the phrase has been replaced.

  • Line 424: What is the “balance of relative humidity…”?

The phrase has been simplified, but the meaning has not changed.

  • Line 457: “author’s” should be “authors’”

Replaced "author’s" with "authors".

  • Line 470: “This diagram of mineralogical…” might be better. What does the word “objects” refer to?

The proposal was rewritten.

  • Line 488: A bit unclear. Are you saying that the gypsum and halite content increases from the periphery toward the center of the reservoir?

That's right, the proposal was corrected.

  • Line 488: Any place in the manuscript that you use the term “reservoir”, change it to “lake”. If the lake is actually sometimes used as a reservoir, ignore this comment.

Throughout the text, the term "reservoir" was replaced by "lake".

  • Line 499: “objects” is probably the wrong word. How about “features”?

Replaced "objects" with "features".

Mineral Nomenclature (Table 1, Table 3 and elsewhere)

  • Chlorite is a mineral group, not a species—a footnote about this would be good.

Replaced "chlorite" with "minerals of the chlorite group

  • Thenardite should be Thénardite.

Replaced " Thenardite " with "Thénardite".

  • Astrakhonite should be Blödite.

Replaced "Astrakhonite" with "Blödite".

  • Gibbsit should be Gibbsite.

Replaced "Gibbsit" with "Gibbsite".

  • Chalcedony is a cryptocrystalline variety of Quartz, not a separate mineral species

Replaced "Chalcedony" with "quartz (cryptocrystalline variety)".

  • Nontronite has a nonstandard chemical formula listed. I think it should be Na3Fe3+2(Si,Al)4O10(OH)2.nH2O.

The nontronite formula has been corrected.

  • The seven Montmorillonite listings in table 3 are confusing. Fe-Mg Montmorillonite is not a mineral species, you probably mean a ferroan, magnesian Montmorillonite (or a ferrian, magnesian montmorillonite—I can’t tell). Why are there two listings for just  Montmorillonite.  All the other listings are chemical variants and not recognized minerals.  Maybe a footnote explaining what you are trying to do would be useful.

In the table, seven varieties of montmorillonite are combined into one group of montmorillonite. The second column of the table shows the formulas of montmorillonite that were used in the calculations.

  • The three chemical variants of Beidellite are also confusing. There is only one recognized mineral species, Beidellite. The rest are magnesian Beidellite, calcian Beidellite, and sodic Beidellite.

Similarly, all Beidellites used in the calculations were combined in the table.

  • Illite is a commonly used geological term, but there is no IMA-approved mineral species, Illite. Also, how can K-illite have less potassium in it than plain old Illite?

We suggest leaving the term "illite" in this work. It is adopted by the Nomenclature Committee for Clay Minerals and implies the group name for all micaceous minerals. A reference to the recommendations of the Nomenclature Committee, as well as an explanation of this term, is given in clause 4.1.1. of this article. In the table 3, "K-illite" was replaced by "Illite".

  • Why use Anorthite for the calcium end-member of the plagioclase feldspar series, but not Albite for the sodium end-member?

Corrected. The sodium end-member of the plagioclase feldspar series – albite.

  • Melanterite is a ferrous sulfate, whereas Jarosite is a ferric sulfate. The formulas should indicate this.

They believed the formulas of melanterite and jarosite. Formulas are correct.

  • I don’t believe Staccite is a valid mineral species. A reference for this term is needed.

An incorrect conversion of the mineral species has occurred. Replaced "Staccite" with "Scacchite".

Is it possible to left-justify the list of minerals in table 3?  The variable indentations from having the text centered imply significance to the apparent indentation, but there isn’t any!

The list of minerals in Table 3 is left aligned.

Technical Issues

  • Scattered throughout the manuscript sodium ions are listed as divalent, i.e. Na2+. This is wrong. Sodium is monovalent, i.e. Na+.  These must be corrected.

The sodium valence was corrected throughout the text.

  • Lines 442-446: I think the equations shown should be balanced. I don’t know what O In equation 2, there are five oxygen atoms on the left, but eight oxygen atoms on the right…?

Equation (2) was corrected. O3 – ozone in the surface air. It is assumed that surface ozone concentration is up to few tens of µg/m-3 at cloudless weather and up to few hundreds of µg/m-3 during precipitation.

  • In the paragraph starting at Line 288, I think the type-setting of HCO3is garbled. The carbonic acid sequence in question is H2CO3, HCO3-, CO32-.

Didn't quite understand the question. Corrected the sequence in the "Materials and Methods" section.

Round 2

Reviewer 1 Report

The article can be accepted. But it should be recommended to include the references on the literatures devoted to mineralogy of high temperature minerals like cristobalite, tridymite occurring in the lake deposits. 

Author Response

Thanks to the reviewer for the necessary additions. The link to the publication has been added. In the text of the article, the link is marked in blue.
